# Preparation and Physicochemical Characterization of Water-Soluble Pyrazole-Based Nanoparticles by Dendrimer Encapsulation of an Insoluble Bioactive Pyrazole Derivative

**DOI:** 10.3390/nano11102662

**Published:** 2021-10-10

**Authors:** Silvana Alfei, Chiara Brullo, Debora Caviglia, Guendalina Zuccari

**Affiliations:** 1Department of Pharmacy, University of Genoa, Viale Cembrano, 16148 Genoa, Italy; brullo@difar.unige.it (C.B.); zuccari@difar.unige.it (G.Z.); 2Department of Surgical Sciences and Integrated Diagnostics (DISC), University of Genoa, Viale Benedetto XV 6, 16132 Genoa, Italy; Cavigliad86@gmail.com

**Keywords:** fourth-generation polyester-based lysine-modified dendrimer, physical encapsulation, 2-(4-Bromo-3,5-diphenyl-pyrazol-1-yl)-ethanol (BBB4), water-soluble BBB4-loaded NPs, spherical morphology, biphasic release profile, first-order kinetics, high drug loading, NMR investigations

## Abstract

2-(4-Bromo-3,5-diphenyl-pyrazol-1-yl)-ethanol (BBB4) was synthetized and successfully evaluated concerning numerous biological activities, except for antimicrobial and cytotoxic effects. Due to the antimicrobial effects possessed by pyrazole nucleus, which have been widely reported, and the worldwide need for new antimicrobial agents, we thought it would be interesting to test BBB4 and to evaluate its possible antibacterial effects. Nevertheless, since it is water-insoluble, the future clinical application of BBB4 will remain utopic unless water-soluble BBB4 formulations are developed. To this end, before implementing biological evaluations, BBB4 was herein re-synthetized and characterized, and a new water-soluble BBB4-based nano-formulation was developed by its physical entrapment in a biodegradable non-cytotoxic cationic dendrimer (G4K), without recovering harmful solvents as DMSO or surfactants. The obtained BBB4 nanoparticles (BBB4-G4K NPs) showed good drug loading (DL%), satisfying encapsulation efficiency (EE%), and a biphasic quantitative release profile governed by first-order kinetics after 24 h. Additionally, BBB4-G4K was characterized by ATR-FTIR spectroscopy, NMR, SEM, dynamic light scattering analysis (DLS), and potentiometric titration experiments. While, before the nanotechnological manipulation, BBB4 was completely water-insoluble, in the form of BBB4-G4K NPs, its water-solubility resulted in being 105-fold higher than that of the pristine form, thus establishing the feasibility of its clinical application.

## 1. Introduction

The five membered heterocyclic diazole ring of pyrazole and its derivatives represent versatile template structures for designing potent bioactive agents [1,2,3]. In fact, molecules belonging to the pyrazole family have demonstrated possessing numerous biological activities, such as antimicrobial, anti-inflammatory, anticancer, analgesic, anticonvulsant, anthelmintic, antioxidant, and herbicidal effects, mainly due to the planar structure of the aromatic heterocycle [1,2,3]. In particular, the structure of the pyrazole ring presents three carbon atoms and two adjacent nitrogen atoms, and the pyrazole nucleus is also present in natural compounds (although rarely, probably due to difficulty in the formation of the N=N bond by living organisms) [4]. Regarding this, the naturally occurring amino acid *L*-α-amino-β-(pyrazolyl-N)-propanoic acid [(S)-β-pyrazolyl alanine], isolated in 1957 from the *Citrullus vulgarisin* juice of watermelon, was the first example of a pyrazole containing natural product, endowed with anti-diabetic activity.

Due to their several biological activities in humans, pyrazole derivatives are thought to be alkaloids [1]. The pyrazole nucleus possesses almost all types of pharmacological activities and, throughout the years, has attracted the attention of many researchers, who have studied its skeleton both chemically and biologically. Currently, several studies have been reported on the synthesis and biological activity of pyrazole derivatives worldwide [2]. Currently, the pyrazole nucleus is present in numerous pharmacological agents belonging to different therapeutic categories, including Celecoxib (potent anti-inflammatory by COX-2 inhibition), Tepoxalin (nonsteroidal anti-inflammatory drugs (NSAIDs)), Crizotimib (anticancer), Surinabant, Difenamizole (anti-obesity), Mepiprazole (tranquillizer), Finopril (insecticide), CDPPB (antipsychotic), Betazole (analgesic), and Fezolamide (H2-receptor agonist and antidepressant agent), confirming the pharmacological potential of the pyrazole nucleus [2,3]. Figure 1 reports some of the marketed and well-established therapeutics containing the pyrazole unit.

To confirm this scenario, several previously synthetized 3,5-diphenylpyrazole derivatives, including 2-(4-bromo-3,5-diphenyl-pyrazol-1-yl)-ethanol (BBB4, Figure 2), showed several pharmacological activities [5]. In particular, BBB4 showed analgesic, hypotensive, anti-inflammatory, local anaesthetic, and motor activity inhibition effects in mice and rats, while showing mild platelet antiaggregating action in vitro [5].

On the contrary, the activity of BBB4 as a cytotoxic and/or antimicrobial agent was not investigated, since the first studies reporting the antimicrobial effects and the cytotoxic action of pyrazole derivatives have been published more recently [6,7]. Consequently, its possible antibacterial activity and cytotoxic behaviour towards both cancer and normal eukaryotic cells, to date, remain unveiled and need to be investigated.

Additionally, the bacterial infections caused by bacteria of the ESKAPE (Enterococcus faecium, Staphylococcus aureus, Klebsiella pneumonia, Acinetobacter baumannii, Pseudomonas aeruginosa, Enterobacter) family, which are recognized as dangerous superbugs by the Infectious Diseases Society of America (IDSA), are becoming a global concern because they are almost untreatable. ESKAPE bacteria are endowed with an increasing tendency to develop multi-drug resistance (MDR) to the bactericidal effect of conventional antibiotics, which are no longer effective and need substitution with more efficient new antibacterial agents suitable for clinical application.

In this regard, since the antimicrobial properties of molecules containing the pyrazole nucleus are now extensively documented [3], BBB4 could be a suitable candidate to develop a novel antibacterial agent. Unfortunately, even if it were found to be capable of inhibiting bacteria in vitro, due to its water-insolubility, its possible clinical application would remain utopic unless water-soluble BBB4 formulations are developed. Consequently, before performing biological evaluations of BBB4, we believed it to be essential to strive toward making BBB4 water soluble.

To solubilize drugs without using harmful solvents or high amounts of surfactants and emulsifiers, often responsible for adverse reactions in patients, the most consolidated strategies consist of using nanosized reservoirs such as liposomes [8], hyperbranched polymers [9,10], or dendrimers [11].

To our knowledge, only two studies exist in the literature concerning the nano-encapsulation of two out of the several reported bioactive pyrazole derivatives poorly soluble in water [12,13], but only one concerns a nanotechnology application to enhance water solubility [13]. In particular, Sun et al. [13] have recently encapsulated the pyrazole derivative 6-amino-4-(2-hydroxyphenyl)-3-methyl-1,4-dihydropyrano [2,3-c] pyrazole-5-carbonitrile (AMDPC) in commercial polymers, poly (ethylene glycol) methyl ether-block-poly(lactide-co-glycolide) (PEG-PLGA), obtaining micelles which gave clear water solutions at 0.05 mg/mL, while at the same concentration, pristine AMDPC was insoluble. Although AMDPC was turned water soluble, considering the very low DL% (1.28) obtained in this study, the water solubility obtained for the encapsulated AMDPC was insignificant (0.64 × 10^−4^ mg/mL).

Dendrimers could help in improving these results. Dendrimers, unlike traditional polymers, are three dimensional macromolecules with a unique tree-like branching architecture and globular shape. Dendrimers peripherally cationic of high generation are highly water-soluble, due to numerous peripheral hydrophilic groups compatible with water. In addition, they possess inner hydrophobic cavities for hosting lipophilic molecules in high, thus allowing for high DL% values and high water-solubility improvements.

Given this scenario, aiming at solving the solubility drawbacks of BBB4, which contradict its applicability as therapeutic and thus nullify its potentials as medicinal, in this study, we used a lysine-containing fourth generation biodegradable dendrimer synthetized by us, which was not cytotoxic, to encapsulate and solubilize BBB4 in water. The new BBB4-based formulation showed high drug loading (DL%), good encapsulation efficiency (EE%), and a biphasic quantitative release profile that fitted the first-order kinetic model. Attenuated total reflection-Fourier transform infrared (ATR-FTIR) spectroscopy, nuclear magnetic resonance (NMR) analysis, dynamic light scattering (DLS) studies, and potentiometric titrations were also performed to complete the characterization of the obtained BBB4-dendrimer formulation. Principal component analysis (PCA) was exploited to process FTIR spectral data of G4K, BBB4, and BBB4-G4K, to obtain reliable information concerning BBB4-G4K’s chemical composition.

## 2. Material and Methods

### 2.1. Chemicals and Instruments

The dendrimer utilized for encapsulating the pyrazole derivative BBB4, namely G4K, was prepared starting from the *bis*-hydroxymethyl propanoic acid (*bis*-HMPA), following a multi-step procedure [11,14,15,16,17,18] and schematized in Section 3.1. The ATR-FTIR and NMR data, as well as the elemental analysis results of the main intermediates (the uncharged inner scaffold (G4OH) and the *tert*-butyloxycarbonyl (Boc)-protected lysine dendrimer (G4BK)) and of the final lysine dendrimer hydrochloride salt (G4K) have been reported in Section 2.2.1, Section 2.2.2, and Section 2.2.3, respectively. BBB4 was prepared according to the procedure schematized in Section 3.2 [5]. The ATR-FTIR and NMR data, as well as the elemental analysis results, confirmed its structure and purity and have been reported in Section 2.2.4. Methyl alcohol (MeOH) and acetonitrile for HPLC analyses were HPLC grade and were obtained from Merck (formerly Sigma-Aldrich, Darmstadt, Germany). All reagents and solvents were purchased from Merck (formerly Sigma-Aldrich, Darmstadt, Germany) and were reagent grade. Solvents were purified by standard procedures, whereas reagents were employed as such, without further purification. Melting points and boiling points are uncorrected. ^1^H and ^13^C NMR spectra of all compounds were acquired on a Jeol 400 MHz spectrometer (JEOL USA, Inc., Peabody, MA, USA) at 400 and 100 MHz, respectively. Fully decoupled ^13^C NMR spectra were reported. Chemical shifts were reported in ppm (parts per million) units relative to the internal standard tetramethylsilane (TMS = 0.00 ppm), and the splitting patterns were described as follows: s (singlet), d (doublet), t (triplet), q (quartet), m (multiplet), and br (broad signal). Centrifugations were performed on an ALC 4236-V1D centrifuge at 3400–3500 rpm. Elemental analyses were performed on an EA1110 Elemental Analyser (Fison Instruments Ltd., Farnborough, Hampshire, UK). Column chromatography was performed using Merck (Washington, DC, USA) silica gel (70–230 mesh) as stationary phase. Scanning electron microscopy (SEM) images were obtained with a Leo Stereoscan 440 instrument (LEO Electron Microscopy Inc., Thornwood, NY, USA). Dynamic Light Scattering (DLS) and Z-potential (ζ-*p*) determinations were performed using a Malvern Nano ZS90 light scattering apparatus (MalvernInstruments Ltd., Worcestershire, UK). Potentiometric titrations were performed with a Hanna Micro-processor Bench pH Meter (Hanna Instruments Italia srl, Ronchi di Villafranca Padovana, Padova, Italy). Lyophilization was performed using a freeze–dry system (Labconco, Kansas City, MI, USA). A thin layer chromatography (TLC) system employed aluminium-backed silica gel plates (Merck DC-Alufolien Kieselgel 60 F254, Merck, Washington, DC, USA), and detection of spots was made by UV light (254 nm) using a handheld UV lamp, LW/SW, 6W, UVGL-58 (Science Company^®^, Lakewood, CO, USA). Organic solutions were dried over anhydrous magnesium sulphate and were evaporated using a rotatory evaporator operating at a reduced pressure of about 10–20 mmHg.

### 2.2. ATR-FTIR, NMR, and Elemental Analysis of G4OH, G4BK, G4K, and BBB4

#### 2.2.1. G4OH

Fluffy white solid (98% isolated yield), m.p. 77 °C. FTIR (ν, cm^−1^): 3424 (OH), 1739 (C=O). ^1^H NMR (DMSO*-d6*, 400 MHz): δ = 0.80 (s, 3H, CH_3_ of *core*), 1.01 (s, 72H, CH_3_ of fourth generation (G4)), 1.16 (s, 36H, CH_3_ of third generation (G3)), 1.18 (s, 18H, CH_3_ of second generation (G2)), 1.22 (s, 9H, CH_3_ of first generation (G1)), 3.29–3.49 (m, 96H, CH_2_OH); 4.08–4.30 (m, 90H, CH_2_O of dendrimer), 4.55 (br q, 48H, OH). ^13^C NMR (DMSO*-d6*, 100 MHz): δ = 16.67, 16.84, 16.88, and 17.12 (CH_3_), 46.16, 46.19, 46.23 and 50.20 (quaternary C), 63.63 (CH_2_OH), 64.33, 64.86 and 65.29 (CH_2_O), 171.42 (two signals overlapped), 171.79 and 174.00 (C=O), CH_3_, quaternary C and CH_2_O of *core* were no detectable. Anal. Cald. for C_230_H_372_O_138_: C, 51.68; H, 7.01%. Found: C, 51.86; H 7.18.

#### 2.2.2. G4BK

Viscous resin (90% isolated yield). FTIR (ν, cm^−1^): 3431 (NH), 1739 (C=O ester), 1694 (C=O urethane), 1528 (NH). ^1^H NMR (CDCl_3_, 400 MHz): δ ≤ 1.00 (CH_3_ of *core* was no detectable), 1.00–1.70 (m, 423 H, CH_3_ of G1, G2, G3, G4 + CH_2_CH_2_CH_2_ of lys), 1.36 (s, 432 H, CH_3_ of Boc), 1.37 (s, 432 H, CH_3_ of Boc), 2.87 (m, 96 H, CH_2_NH), 3.44–4.22 (m, 234 H, CH_2_O of dendrimer + CHNH of lys), 6.65, 6.75 and 7.05 (br signals, 48H, ^ε^NH), 6.95 (d, *J* = 7.9 Hz, 48 H, ^α^NH). ^13^C NMR (CDCl_3_, 100 MHz): δ = 17.00–18.00 (CH_3_ of G1, G2, G3, G4), 22.61 (CH_2_), 28.39 (CH_3_ of Boc), 28.49 (CH_3_ of Boc), 29.58 (CH_2_), 31.81 (CH_2_), 40.06 (CH_2_NH), 46.00–49.00 (quaternary C G1, G2, G3, G4), 53.41 (CHNH), 65.31–65.37 (CH_2_O of G1, G2, G3, G4), 78.99 (quaternary C of Boc), 79.77 (quaternary C of Boc), 155.66 (C=O urethane), 156.23 (C=O urethane), 171.80–172.45 (C=O amino acid + C=O ester of G1, G2, G3, G4), CH_3_, quaternary C and CH_2_O of *core* were not detectable. Anal. Cald. for C_998_H_1716_N_96_O_378_: C, 56.79; H, 8.19; N, 6.37%. Found: C, 56.98; H, 8.49; N, 6.06

#### 2.2.3. G4K

Very hygroscopic glassy solid (92% isolated yield). FTIR (ν, cm^−1^): 3500–3000 (NH_3_^+^), 2930 (alkyl), 1733 (C=O), 1216, 1048 (C-O). ^1^H NMR (DMSO-*d6*, 400 MHz): δ ≤ 1.00 (CH_3_ of *core* was no detectable), 1.03–1.99 (m, 423 H, CH_3_ of G1, G2, G3, G4 + CH_2_CH_2_CH_2_ of lys), 2.76 (m, 96 H, CH_2_NH_3_^+^ of lys), 3.99 (m, 48 H, CHNH_3_^+^ of lys), 4.10–4.50 (m, 186 H, CH_2_O of dendrimer), 8.20 (br s, 144 H, NH_3_^+^ of lys), 8.82 (br s, 144 H, NH_3_^+^ of lys). ^13^C NMR (DMSO-*d6*, 100 MHz): δ = 19.33 (CH_3_), 23.14 (CH_2_), 28.01 (CH_2_), 31.01 (CH_2_), 40.02 (CH_2_NH_3_^+^ of lys), 47.70 (quaternary C), 53.55 (CHNH_3_^+^ of lys), 67.65–67.82 (CH_2_O and of G1, G2, G3, G4), 170.68–173.33 (C=O of amino acid + ester of G1, G2, G3, G4), CH_3_, quaternary C and CH_2_O of core were not detectable. Anal. Cald. for C_518_H_1044_N_96_O_186_Cl_96_: C, 41.48; H, 7.02; N, 8.97; Cl, 22.69%. Found: C, 41.88; H, 7.29; N, 8.86; Cl, 22.21.

#### 2.2.4. BBB4

White solid (90% isolated yield). M.p.: 76–90 °C (Dyethyl ether: Petrol Ether 1:1). FTIR (ν, cm^−1^): 3246 (OH), 3064 (aromatics), 2961, 2908, 2842 (alkyl), 692 (C-Br). ^1^H NMR (CDCl_3_, 400 MHz): δ = 2.70 (br s, 1H, OH), 3.96–4.03 (m, 2 H, CH_2_), 4.16–4.25 (m, 2 H, CH_2_), 7.35–7.57 (m, 8 H, phenyl rings), 7.90–7.98 (m, 2 H, phenyl rings) Anal. Cald. for C_17_H_15_BrN_2_O: C, 59.49; H, 4.41; N, 8.16%. Found: C, 59.14; H, 4.36; N, 8.41.

### 2.3. Cytotoxicity Studies

Dose-dependent studies of cytotoxicity were performed for the reservoir dendrimer G4K. The cytotoxicity of G4K was evaluated in vitro on HeLa cell lines purchased by Thermo Fischer Scientific (Rodano, Milan, Italy). Briefly, HeLa cells were increased in Dulbecco’s Modified Eagle Medium (DMEM) enriched with foetal bovine serum (FBS, 10%), non-essential amino acids (1%), and antibiotics (1%, penicillin and streptomycin) and maintained in an atmosphere containing 5% CO_2_ at 37 °C. The cells were seeded at a density of 2 × 10^4^ cells per well in a 24-well plate and in 4-well slides in 500 μL of medium and incubated at 37 °C for 72 h. Subsequently, the cells were incubated with increasing concentrations (1–100 µM) of G4K at 37 °C for 24 h. Then, 10 µL MTT [3-(4,5-dimethylthiazol-2-yl)-2,5-diphenyl-2H-tetrazolium bromide] was added into each well, and after 4 h, the medium and MTT were discarded and 100 µL dimethyl sulfoxide (DMSO) was added into each well. Finally, optical density at 490 nm was measured on a Thermo Fischer Scientific microplate reader (Rodano, Milan, Italy) to determine cell viability (%). Paclitaxel was assayed in the same condition as a positive control. Determinations were made in triplicate, and results were expressed as mean percentage of the control (untreated cells) ± standard deviation (SD).

### 2.4. Experimental Procedure to Prepare BBB4-Loaded Dendrimer NPs (BBB4-G4K)

The BBB4-G4K NPs were prepared according to a reported procedure modified as follows [19]. A total of 93.5 mg (6.2 μmol) of G4K dendrimer (Gen 4.0) was dissolved in 7.5 mL of deionized water (pH = 7.4). To the dendrimer water solution, a strong excess of BBB4 (98.0 mg) and a total of 6 mL of ethanol (EtOH) in three aliquots to achieve a clear solution were added. The resulting solution was incubated for 48 h at 37 °C under vigorous stirring, and following incubation, the solution was evaporated. The evaporation of the hydroalcoholic mixture was performed using a Rotavapor^®^ R-3000 (Büchi Labortechnik, Flawil, St. Gallen, Switzerland) at 100 °C and reduced pressure. The waxy solid residue was washed under stirring for 1 h with acetone to extract the BBB4 not encapsulated. The acetone washings were separated and the BBB4-loaded dendrimer nanoparticles (BBB4-G4K NPs), purified by the free UA, were obtained as an orange, sticky solid which was stored under vacuum in a dryer (132.6 mg). Non-encapsulated BBB4 was recovered by evaporating the acetonic solution and was obtained in the form of an off-white solid (64.0 mg). The solid was recrystallized from dyethyl ether:petrol ether 1:1, and its structure was confirmed by ATR-FTIR analysis (spectrum not reported).

#### Spectroscopic Data Related to BBB4-G4K NPs

FTIR (ν, cm^−1^): 3500–3000 (NH_3_^+^ dendrimer, OH stretching BBB4), 2985, 2880 (alkyl groups of dendrimer and BBB4), 1736 (C=O stretching esters of dendrimer), 1220, 1051 (C-O stretching esters of dendrimer), 697 (C-Br stretching of BBB4).

^1^H NMR (CD_3_OD, 400 MHz): δ ≤ 1 (CH_3_ *core* not detected), 1.00–2.00 [m, 135 H (CH_3_ G1, G2, G3 and G4 of dendrimer) + 288 H (CH_2_CH_2_CH_2_ of lys)], 2.95–3.16 [m, 96H (CH_2_NH_3_^+^ of lys)], 3.96–4.03 [m, 36H (CH_2_ of BBB4)], 3.99 (m, 48 H, CHNH_3_^+^ of lys), 4.16–4.25 [m, 36 H, (CH_2_ of BBB4)], 4.30–4.50 [m, 186 H (CH_2_O of dendrimer), 7.35–7.57 [m, 144 H (phenyl rings of BBB4)], 7.90–7.98 [m, 36 H (phenyl rings of BBB4)], NH_3_^+^ of lys (288 H) and OH of BBB4 (18 H) were not detectable because the protons from these groups are exchanged with the proton of CD_3_OD. From ^1^H NMR analysis: C_824_H_1314_N_132_O_204_Cl_96_Br_18_; MW = 21,175.8.

### 2.5. Chemometric Assisted ATR-FTIR Spectroscopy

FTIR spectra of BBB4-G4K, BBB4, and G4K were recorded in triplicate directly on the solid samples in attenuated total reflection (ATR) mode using a Spectrum Two FTIR Spectrometer (PerkinElmer, Inc., Waltham, MA, USA). Acquisitions were made from 4000 to 600 cm^−1^, with 1 cm^−1^ spectral resolution, co-adding 32 interferograms, with a measurement accuracy in the frequency data at each measured point of 0.01 cm^−1^, due to the laser internal reference of the instrument. The frequency of each band was obtained automatically by using the “find peaks” command of the instrument software. The matrix of spectral data was subjected to PCA using PAST statistical software, (paleontological statistics software package for education and data analysis, freely downloadable online at: https://past.en.lo4d.com/windows; accessed on 13 September 2021). We organized the FTIR data sets of the spectra acquired for BBB4, G4K, and BBB4-G4K in a matrix of *n* = 10,203 measurable variables. For each sample, the variables consisted of the values of absorbance (%) associated with the wavenumbers (3401) in the range 4000–600 cm^−1^. The system was simplified exploiting the PCA, which is a chemometric tool able to reduce the large number of variables to a small number of new variables, namely principal components (PCs). 

### 2.6. Morphology of Particles of G4K and BBB4-G4K

The morphology of G4K and BBB4-G4K was investigated by scanning electron microscopy (SEM). In the performed experiments, samples were fixed on aluminium pin stubs and sputter-coated with a gold layer of 30 mA for 1 min, and an accelerating voltage of 20 kV was used for the sample’s examination. The micrographs were recorded digitally using the DISS 5 digital image acquisition system (Point Electronic GmbH, Halle, Germany).

### 2.7. Content of BBB4 in BBB4-G4K NPs, Drug Loading (DL%), and Encapsulation Efficiency (EE%)

#### 2.7.1. BBB4 Calibration Curve

An initial solution of BBB4 (1 mg/mL) was prepared in MeOH, and dilutions with MeOH were made to prepare standard solutions at concentrations of 50, 100, 200, 300, 400, and 500 µg/mL. Aliquots of 20 µL were picked up from each solution and were analysed to construct the BBB4 standard calibration curve. In particular, BBB4 in each solution was quantified using an HPLC JASCO system (Jasco Inc., Easton, MD, USA) equipped with a JASCO PU-980 pump, a JASCO UV-970-975 UV/Vis detector, and an ODS C18 column (250 × 4.6 mm, 5 µm), by detecting the absorbance (A) at ʎ_abs_ = 253 nm. The mobile phase consisted of a mixture of acetonitrile and 10 mM K_2_HPO_4_ aqueous buffer solution (15/85, *v*/*v*). The column was preconditioned for at least 20 min before the first injection. The running time was set at 20 min. The BBB4 calibration curve was obtained by a least-squares linear regression analysis of the BBB4 concentrations vs. the A signals created in the UV detector by the different concentrations of analyte (BBB4). Determinations were made in triplicate, and the A values obtained for each BBB4 concentration analysed were expressed as mean ± standard deviation (A_mean_ ± SD).

#### 2.7.2. Estimation of BBB4 Content in BBB4-G4K NPs 

A total of 5 mg of BBB4-G4K was dissolved in 10 mL of MeOH (500 µg/mL), and the clear solution was vigorously stirred for ten minutes to promote the release of BBB4. The amount of BBB4 in the sample was quantified at ʎ_abs_ = 253 nm by HPLC analysis, using the same apparatus and the same conditions described in the previous section. In particular, six aliquots (20 µL) of the solution were analysed against a blank solution of the empty dendrimer G4K. 

The values of DL% and EE% of BBB4-G4K were calculated from the following Equations (1) and (2):(1)$$DL\;\left(\%\right)=\frac{\mathrm{weight}\;\mathrm{of}\;\mathrm{the}\;\mathrm{drug}\;\mathrm{in}\;\mathrm{NPs}}{\mathrm{weight}\;\mathrm{of}\;\mathrm{the}\;\mathrm{NPs}}\times 100\;\;\;\;\;$$
(2)$$EE\;\left(\%\right)=\frac{\mathrm{weight}\;\mathrm{of}\;\mathrm{the}\;\mathrm{drug}\;\mathrm{in}\;\mathrm{NPs}}{\mathrm{inizial}\;\mathrm{amount}\;\mathrm{of}\;\mathrm{drug}}\times 100\;$$

### 2.8. Molecular Weight of BBB4-G4K NPs

According to a previously reported procedure [11], the MW of BBB4-G4K was estimated both by ^1^H NMR analysis and by HPLC analyses, obtaining results with a minimal difference (0.5%).

### 2.9. Water Solubility of BBB4, UA-G4K NPs, and the Nanotechnologically Manipulated UA Released in Water

The water solubility of pristine BBB4, BBB4 in the form of BBB4-G4K NPs, and nanoengineered BBB4 solubilized in water was determined with the shake-flask method [20]. An exactly weighted excess of BBB4 and BBB4-G4K (13.3 mg) was added with water m-Q (1.5 mL), obtaining suspensions which were maintained under vigorous stirring at room temperature, observing, for BBB4-G4K only, abundant foaming (pH = 7.4). The suspensions were stirred until producing an equilibrium between the saturated solution and undissolved BBB4 and BBB4-G4K. Then, the suspensions were centrifugated (15 min, 3500 rpm) to remove undissolved BBB4 and BBB4-G4K and drops of the supernatant solutions were observed with a Leica Galen III Professional Microscope (Taylor Scientific, St. Louis, MO, USA), without observing precipitate or differences with a drop of pure water. The solid residues were washed twice with acetone to help water removal and brought to constant weight under vacuum, obtaining, in the case of pristine BBB4, 13.24 ± 0.03 mg of insoluble material, while 3.70 ± 0.08 mg of residual was obtained in the case of BBB4-G4K NPs. The amounts of BBB4 (0.06 ± 0.03 mg) and of BBB4-G4K (9.6 ± 0.08 mg) solubilized in water were obtained for difference from the initial amount added in 1.5 mL water (13.3 mg). The experiments were made in triplicate, and the solubilities of pristine BBB4 and of BBB4 in the form of BBB4-G4K NPs were reported as mean ± SD. Additionally, the water solution obtained by dissolving BBB4-G4K was diluted to have a final volume of 10 mL with MeOH, and 20 µL aliquots were analysed by HPLC using the same apparatus and the same conditions described in Section 2.7.1. The exact amount of BBB4 which was solubilized in water was quantified at 253 nm using the previously constructed standard calibration curve. The determinations were made in triplicate, and the BBB4 water solubility was reported as mean ± SD.

### 2.10. Dynamic Light Scattering (DLS) Analysis

Particle size (in nm), polydispersity index (PDI), and zeta potential (ζ-*p*) (mV) of BBB4-G4K were measured at 25 °C at a scattering angle of 90° in m-Q water by using a Malvern Nano ZS90 light scattering apparatus (Malvern Instruments Ltd., Worcestershire, UK).

Solutions of BBB4-G4K in m-Q water were diluted to final concentrations to have 250–600 kcps. ζ-*p* value of BBB4-G4K was recorded with the same apparatus. The results from these experiments were presented as the mean of three different determinations ± SD. Concerning the particle size distribution, intensity-based results were reported.

### 2.11. Potentiometric Titration of G4K and BBB4-G4K

Potentiometric titrations were performed at room temperature to construct the titration curves of G4K and BBB4-G4K. The samples (20–30 mg) were dissolved in 30 mL of Milli-Q water (m-Q) and then were treated with a standard 0.1 N NaOH aqueous solution (1.5 mL, pH = 9.34 (G4K) and 9.54 (BBB4-G4K)). The solutions were potentiometrically titrated by adding 0.2 mL aliquots of a standard 0.1 N HCl aqueous solution, up to a total of 3.0 mL, and measuring the corresponding pH values [21]. Titrations were made in triplicate, and the determinations were reported as mean ± SD.

### 2.12. In Vitro BBB4 Release Profile from BBB4-G4K NPs

In vitro release of BBB4 from BBB4-G4K NPs was investigated using the dialysis bag diffusion method. An exactly weighted amount of BBB4-G4K (10 mg) was dissolved in 2 mL of 0.1 M phosphate-buffered saline (PBS, pH = 7.4), which should assure the dissolution of the complex. The solution was then placed into a pre-swelled T2 tubular cellulose dialysis bag (flat width = 10 mm, wall thickness = 28 µm, V/cm = 0.32 mL) with a nominal molecular weight cut-off (MWCO) of 6000–8000 Da (Membrane Filtration Products, Inc., Seguin, TX, USA) and dipped into 20 mL of 0.1 M PBS, pH 7.4, at 37 °C with gentle stirring for 24 h. At predetermined time intervals (1 h, 2 h, 3 h, 4 h, 5 h, 6 h, 8 h, 10 h, 12 h, 24 h), 1 mL was withdrawn from the incubation medium and was analysed by HPLC using the same apparatus and the same conditions described in Section 2.7.1 to determine the BBB4 concentration. The exact amount of BBB4 present in the samples was quantified at 253 nm, and the results were reported as the mean ± SD of three determinations. After sampling, an equal volume of fresh PBS was immediately replaced into the incubation medium.

The concentration of BBB4 released from BBB4-G4K NPs was expressed as a cumulative release percentage (%) of the total amount of BBB4 present in the BBB4-G4K NPs (according to the DL% value). The BBB4 cumulative releases (%) were plotted as a function of time obtaining the curve of BBB4 release profile.

### 2.13. Statistical Analysis

The statistical significance of the slope of the BBB4 calibration curve was investigated through the analysis of variance (ANOVA), performing the Fischer test. Statistical significance was established at the *p*-value < 0.05.

## 3. Results and Discussion

### 3.1. Main Dendrimer Intermediates (G4OH, G4BK) and Cationic Dendrimer G4K

Starting from the *bis*-hydroxymethyl propanoic acid (*bis*-HMPA) and according to Figure 1, the multi-step synthesis of the intermediates G4OH and G4BK and of the lysine dendrimer hydrochloride salt (G4K) was implemented [11,14,15,16,17,18]. FTIR and NMR spectra, as well as the elemental analysis results, were in accordance with those reported in the literature [14,15].

### 3.2. Cytotoxicity Studies

Although the scope of the present study did not include biological evaluations, we retained a mandatory determination of the cytotoxic behaviour of the starting parent dendrimer (G4K) to assess the feasibility of our strategy. Dose-dependent in vitro cytotoxicity of G4K was determined by using the HeLa cell line and performing the MTT assay. In parallel, paclitaxel was tested in the same conditions as the positive control. Figure 3 reports the viability of cells observed at concentrations 1–100 µM of the tested compounds, expressed as a mean percentage of the control (untreated cells, correspondent to G4K concentration of 0 µM) ± SD.

Note that we have prepared BBB4-G4K NPs with the scope herein to obtain a water-soluble BBB4-based formulation endowed with physicochemical properties suitable for a future clinical application and with the future perspective of developing a new cationic antibacterial agent. Given these considerations, we performed experiments for 24 h to have data comparable with the minimum inhibitory concentration (MIC) values that we will obtain in future microbiological tests. As observable in Figure 3, G4K showed no cytotoxicity at all concentrations tested, and viable cells fluctuated between 94.9 and 105.3% in accordance with the viability data previously reported for HeLa exposed to a G4 polyester biodegradable dendrimer based on *b*-HMPA [22]. On the contrary, paclitaxel, adopted as a positive control, was remarkably cytotoxic, showing an LD_50_ under 10 µM, confirming the cytotoxicity data reported in a previous study [23]. Experiments to evaluate of the cytotoxicity of BBB4 on normal cells (human keratinocytes HaCaT) are currently undergoing. In any case, to have an idea of the cytotoxic effects of pyrazole-based compounds, we have herein reported (Table 1) the cytotoxicity data found in the literature of some pyrazole derivatives on different human cancer cell lines such as A549 (lung cancer), HeLa (cervical cancer), MCF-7, and BCAP-37 (breast cancer) and a normal cell line (human embryonic kidney HEK-293) [13,24].

Concerning cancer cell lines, these data evidence that the cytotoxic action of pyrazole derivatives strongly depends on their chemical structure and on the cell line type, HeLa cells being the most susceptible to the action of pyrazole-linked benzotriazole-β-naphthol derivatives. However, considering the cytotoxicity of these compounds toward normal human cells, only 3 compounds out of 30 differently structured pyrazole derivatives (10%) showed slight cytotoxicity (LD_50_ > 46 µM), while for the other 27, no cytotoxicity was detected, thus suggesting a low cytotoxicity of BBB4 on normal eukaryotic cells. 

### 3.3. Synthesis of BBB4

BBB4 was prepared according to a reported procedure schematized in Figure 2. The ATR-FTIR and NMR data, as well as the elemental analysis, confirmed its structure.

### 3.4. Concerning the Encapsulation Method Adopted to Prepare BBB4-Loaded Dendrimer NPs (BBB4-G4K)

Aiming at enhancing BBB4’s water solubility for making it administrable in vivo, BBB4 was re-synthetized and characterized to confirm its structure, and by exploiting nanotechnology, it was entrapped into cationic polyester-based dendrimer NPs containing lysine (G4K). G4K was selected as the solubilizing agent for the following reasons. The hydrolysable and non-cytotoxic inner matrix of the ester type of G4K, which proved to be able to nullify the cytotoxic effects of other hosted molecules [11], regardless the eventual cytotoxicity of BBB4 (to be evaluated), would guarantee a low level of toxicity of the BBB4 nano-formulation. Additionally, the peripheral cationic character of the dendrimer carrier, conferred by lysine, would assure high compatibility with water, and would promote the interaction with the negative components of the bacteria surface, thus favouring the bactericidal activity by damaging their membranes, typical of cationic macromolecules [25,26]. In order to not reproduce our previous work [11], a different encapsulation method was adopted, based on the capability of cationic dendrimers with peripheral amine groups (as G4K) to interact by hydrogen bonds with hydrophobic drugs as BBB4-containing nitrogen ethero atoms, complex them, and consequently solubilize them in water [27]. Accordingly, we performed a modified version of the phase dissolution method reported by Yogesh et al. [19] drafted in the Figure 3.

As the original method, we used water to solubilize the water-soluble dendrimer (12.5 mg/mL) and added EtOH to promote the solubilization of the hydrophobic and water insoluble BBB4 (46 equiv.). Differently from the original method, which required the use of Tween 20 (2.5% *v*/*v*) to help the solubilization of the hydrophobic drug, in our modified procedure, no surfactants or other additives, which in any future clinical use of BBB4-G4K could be responsible for unwanted side reactions, were added. Moreover, a much higher amount of BBB4 was added to the water solution (46 equiv. vs. 19.1 equiv.), and a consequently higher amount of EtOH (80% *v*/*v* vs. 25% *v*/*v*) was included to facilitate the BBB4 solubilization. Ethanol was gradually evaporated during the incubation of the unclosed mixture at 37 °C under stirring for 48 h. After removal of water at reduced pressure, the non-encapsulated BBB4 was removed by the solid residue by acetone washing, obtaining a waxy off-white solid which was brought to constant weight at reduced pressure and stored in a dryer under vacuum on P_2_O_5_. Acetone washes were also evaporated, and the residue was recrystallized by petrol ether/Et_2_O to recover untrapped BBB4. Particularly, BBB4 was obtained as off-white crystals, and its identity was confirmed by ATR-FTIR analysis.

#### 3.4.1. Attenuated Total Reflection Fourier Transform Infrared (ATR-FTIR) Spectroscopy

To assess the success of the encapsulation reaction, we firstly acquired the ATR-FTIR spectra of G4K, BBB4, and purified BBB4-G4K NPs and compared them by simple observation.

Figure 4a shows the spectra of G4K (purple line), BBB4 (green line), and GB4-G4K (dark red line). As expected, the spectra of the encapsulating agent (G4K) and of BBB4-loaded NPs (BBB4-G4K) were very similar, due to the major exposition of the functional groups of the dendrimer external envelop, which gave high absorptions and intense bands, overlapping almost all those of the BBB4 stuck inside.

Observing the region of the spectra inside the fuchsia rectangle and magnified in Figure 4b, the typical very intense band of BBB4 due to the stretching of C-halogen (C-Br, 692 cm^−1^), not observable in the spectrum of G4K, was well detectable in that of BBB4-G4K at 697 cm^−1^, thus confirming the presence of BBB4 inside the structure of the nanocomposite. Note that despite falling in a clean region of the spectrum, the OH stretching band of BBB4 was also not detectable in the BBB4-G4K spectrum, probably due to hydrogen bond interactions with the host dendrimer, which are reported to be primarily responsible for the formation of dendrimer–drug complexes [27].

#### 3.4.2. Principal Component Analysis (PCA) of ATR-FTIR Data

To confirm in a more reliable way the presence of BBB4 in the prepared BBB4-G4K NPs, we applied multivariate analysis (MVA) to the FTIR spectral data using principal component analysis (PCA). PCA is a chemometric method widely used to process spectral data consisting of thousands of variables, which are transformed in a reduced number of new variables called principal components (PCs). PCs represent data variability in two dimensions (PC1 vs. PC2, PC1 vs. PC3, etc.) at a time, which are perpendicular to each other in a score plot [28,29]. In the score plots, the samples under study assume specific positions (scores), forming groups of similar and dissimilar compounds. The position taken by each sample on the selected component can give predictive information on its physicochemical condition. In our case, the reciprocal positions of BBB4, G4K, and BBB4-G4K were obtained and are observable in the plot shown in Figure 5, where PC1 vs. PC2 is reported.

Accordingly, BBB4-G4K was located more distantly from BBB4 than from G4K, indicating that in its structure, the chemical groups of G4K prevail over those of BBB4, confirming what was observed in the ATR-FTIR spectra. Note that BBB4-G4K was positioned at score values different from those of G4K, moving towards BBB4, thus validating the presence of BBB4 in BBB4-G4K. Again, the very small shift of the location of BBB4-G4K towards BBB4 can be explained by assuming that BBB4 was mainly encapsulated and not only adsorbed to the surface of the G4K. Consequently, its functional groups resulted in being hidden in the cavities of G4K, thus absorbing very little in the FTIR analyses and providing very small bands.

#### 3.4.3. ^1^H NMR Analysis

The presence of BBB4 in the structure of BBB4-G4K NPs was definitely established by ^1^H-NMR analysis, which was particularly useful for obtaining quantitative information on the chemical composition of BBB4-G4K, providing the number of moles of BBB4 which have been loaded per mole of G4K. Figure 6 shows a comparison between the ^1^H-NMR spectra of BBB4 (**a**), G4K (**b**), and BBB4-G4K (**c**).

Particularly, in the spectrum of BBB4 (a), while the signal of the OH group was not detectable because it was exchangeable with the residual proton atoms of CD_3_OD, a group of signals in the range of 3.5–4.5 ppm belonging to the diastereotopic proton atoms of the two methylene groups, and a group of signals in the range of 7.0–8.0 ppm given by the proton atoms of the phenyl rings was visible and confirmed the structure of BBB4. Note that having BBB4, only alkyl groups linked to heteroatoms did not possess signals under 3.0 ppm. On the contrary, since G4K does not contain aromatic moieties, its spectrum did not show signals over 5.0 ppm, while it did show broad signals at 3.9–4.5 ppm, 2.76, and 1.03–1.99 ppm (**c**). As evidenced with light blue and dark red rectangles, in the spectrum of BBB4-G4K NPs, both signals typical of BBB4 (red rectangles) and peculiar to G4K (light blue rectangles) were observable, thus confirming the presence of both chemical ingredients in the structure of the complex. To determine the number of BBB4 moles loaded per dendrimer mole (18) and the molecular formula of BBB4-G4K, and to compute its molecular weight (MW), signals belonging to G4K only and to BBB4 only in well separated regions of the spectrum were considered. In this regard, Figure 7 shows the ^1^H NMR spectrum of BBB4-G4K, where the integral values associated with the BBB4-G4K peaks (provided by the instrument) are visible.

In particular, the value of the integral of the signal belonging to the CH_2_NH_3_^+^ of G4K (2.76 ppm, signals number 2, Figure 7), which, as previously established, accounts for 96 proton atoms, was used as a reference value to determine the number of protons fitting the integral value of the aromatic signals of BBB4 (7.0–8.0 ppm, signals number 5 and 6, Figure 7). By dividing this data for the number of aromatic proton atoms existing in a molecule of BBB4 (10), the number of BBB4 moles loaded per dendrimer mole was obtained and was 18. Using this information, it was possible to determine the molecular formula and therefore to calculate the MW of BBB4-G4K, whose value was in accordance with the MW determined using the results of DL% from HPLC analyses (error 0.5%).

Note that the signals of NH_3_^+^ groups of lysine (288H) and those of OH groups of BBB4 (18H) were not detectable because the protons from these groups are exchanged with the protons of CD_3_OD. 

### 3.5. Morphology of Particles of G4K and UA-G4K by SEM

The SEM image of G4K NPs shown in Figure 8a was recently reported [11]. It evidenced a spherical morphology and a particle size of about 300 nm. Concerning BBB4-G4K NPs (Figure 8b), the SEM image established that following the encapsulation of BBB4, the spherical shape was maintained.

Concerning drug/dendrimer NPs, a spherical morphology contributes to providing a high surface area, which typically determines a retention in the circulation system for longer periods and a slow metabolism, which translates to the improved therapeutic effects of the transported drugs [30].

### 3.6. BBB4 Content in BBB4-G4K NPs, DL% and EE%

#### 3.6.1. Results and Discussion Concerning the BBB4 Calibration Curve

Table 2 collects the values of A (expressed as A_mean_ ± SD) determined for each BBB4 concentration injected into the HPLC system, the concentrations of BBB4 (C_BBB4_) used for the HPLC analyses, the BBB4 concentrations predicted by the BBB4 calibration model (C_BBB4p_), the residuals, and the absolute error percentages. 

The absorbance values (A_mean_ ± SD) and C_BBB4_ concentrations (µg/mL) reported in Table 2 were used to develop the BBB4 calibration model by the least squares (LS) method with Equation (3):*y* = 0.0369*x* + 0.0244(3)
where *y* is the absorbance measured at λ_abs_ = 253 nm, and *x* is the BBB4 concentration (C_BBB4_) (µg/mL). Figure 9a shows the obtained linear regression curve.

The high value of the coefficient of determination (R^2^) was 0.9998, which stated the linearity of the calibration. The linearity and sensitivity of the developed calibration model were also evaluated by confirming the statistical significance of its slope through the analysis of variance (ANOVA), performing the Fischer test. Statistical significance was established at the *p*-value < 0.05. Equation (3) was exploited for determining the BBB4 concentrations predicted by the model (C_BBB4p_) for each sample (Table 2) which were reported in a dispersion graph vs. the C_BBB4_ to obtain the regression curve correlating the two sets of data (Figure 9b). R^2^ (0.9998), as well as the value of the correlation coefficient R (0.9999), confirmed a strong correlation between the real and the predicted BBB4 concentrations and the goodness of the model. 

#### 3.6.2. Results Concerning the Determination of BBB4 Concentration in the Prepared BBB4-G4K NPs, DL% and EE%

Six aliquots of a BBB4-G4K solution prepared at a concentration of 500 µg/mL were subjected to HPLC analysis, obtaining six values of A at 253 nm (Table 3). Such data were used to determine the related C_BBB4_ (µg/mL) concentrations by employing Equation (3), whose mean ± SD corresponded to the amount of BBB4 contained in 500 µg/mL of BBB4-G4K NPs. This value allowed for determining DL%, EE%, the moles of BBB4 loaded for the G4K mole, and the MW, which was compared with the value obtained by the ^1^H NMR spectrum (Table 3).

A mean ± SD resulted in being 5.34 ± 0.22, and the average concentration ± SD of BBB4 in the BBB4-G4K analysed resulted in being 144.0 ± 5.9 µg/mL, thus establishing that the total BBB4 loaded in the BBB4-G4K NPs obtained from the encapsulation reaction resulted in being 38.2 ± 1.6 mg. The DL% was 28.8%, while EE% was 39.0. The DL% of BBB4-G4K was very high, thus making possible a great bioactivity at a minimal dosage of the formulation. In fact, the DL% of BBB4-G4K NPs was 22.5-fold higher than that obtained recently by Sun et al. [13], encapsulating the pyrazole derivative 6-amino-4-(2-hydroxyphenyl)-3-methyl-1,4-dihydropyrano[2,3-c]pyrazole-5-carbonitrile (AMDPC) in the commercial polymers poly(ethylene glycol) methyl ether-block-poly(lactide-co-glycolide) (PEG-PLGA), and obtaining micelles with DL% = 1.28. On the contrary, the EE% of BBB4-G4K was lower than that of the micelles [13]. Generally, high values of EE% mean very long times for achieving an only partial release of the entrapped drugs. Since our future perspective is testing BBB4-G4K NPs as a new antibacterial agent, and the antibacterial and bactericidal effects of a molecule are commonly determined at 24 h, a rapid and quantitative drug release of BBB4, more easily allowed by a not too high value of EE%, was desired, and the EE% determined for BBB4-G4K was appreciable.

### 3.7. Determination of UA-G4K Molecular Weight (MW)

The MW of BBB4-G4K was estimated by both its ^1^H NMR spectrum considering the values of integrals of selected peaks as described in Section 3.4.2. and the results of HPLC analyses. A minimal difference of 0.5% was obtained for the results (Table 3, last column), thus confirming the goodness of the BBB4 calibration model and the reliability of the DL% value.

Briefly, in the ^1^H NMR spectrum, a well-separated peak belonging to the CH_2_NH_3_^+^ groups of the peripheral lysine moieties of G4K, whose number of proton atoms is known (96), was considered. By making the proper ratios between the value of the integral of this peak and that of the integral of the peaks due to the aromatic proton atoms, belonging to BBB4 alone, the number of BBB4 moles entrapped in one mole of G4K was determined, which was 18. Then, the MW of UA-G4K was determined according to Equation (4):MW_BBB4-G4K_ = MW of G4K (14,997.9) + 18 × MW of BBB4 (343.2)(4)

The MW of BBB4-G4K was further computed using the results obtained by the HPLC analysis, which established that the moles of BBB4 loaded per G4K mole were 17.7 ± 0.7, according to Equation (5):MW_BBB4-G4K_ = MW of G4K (14,997.9) + 17.7 ± 0.7 x MW of UA (343.2)(5)

Considering the SD value (0.7), it can be noted that the MW estimated by HPLC analysis results can vary from 20,832.3 to 21,312.8, a range that exactly contains the MW value (21,175.8) obtained by ^1^H NMR.

### 3.8. Water Solubility of BBB4-G4K and of Nanotechnologically Manipulated BBB4 Solubilized in Water

The water solubility of BBB4 in the form of BBB4-G4K and that of the nanoengineered BBB4 released in water were determined performing the shake-flask method as described in Section 2 [20]. Table 4 collects the results.

According to a study which reported the experimental determination of the solubility of 16 pharmacologically active substances at pH = 1.2, 4.5, and 6.8 by the shake-flask method and their classification in highly soluble and not highly soluble molecules, at pH = 6.8, which is a pH value similar to that of our experiments, drugs with solubility in the range of 1.73–5.13 mg/mL were classified as highly soluble [31]. Following this classification, considering BBB4 in the form of BBB4-G4K NPs and BBB4 found in a water solution, they can be classified as highly soluble. Considering that pristine BBB4 was practically water-insoluble (0.06 ± 0.03 mg/mL), the water solubility of BBB4-G4K and of BBB4 released in water after nano-manipulation was 105- and 30-fold higher, respectively (Figure 10).

### 3.9. In Vitro Study of BBB4 Release Profile

The release profile of BBB4 from BBB4-G4K NPs was studied by the dialysis method in PBS receptor medium (pH = 7.4). The BBB4 released was determined at fixed points for 24 h, analysing each aliquot in triplicate by HPLC, and the results were expressed as a BBB4 cumulative release percentage (CR%), which is given for each time by Equation (6):(6)$$CR\;\left(\%\right)=\frac{BBB4\left(t\right)}{BBB4\left(NPs\right)}\times 100\;$$
where *BBB*4(*t*) is the amount of BBB4 released at (t) incubation time, while *BBB*4(*NPs*) is the total BBB4 entrapped in the weight of the BBB4-G4K NPs analysed, according to the computed DL%.

CR% values were reported in a dispersion graph vs. the incubation times (Figure 11).

According to a previous study, the release profile of the pyrazole derivative, namely AMDPC, from PEG/PLGA-based micelles was monitored for 48 h [13], while we monitored the BBB4 release for a minor period of 24 h. This different procedure was functional for our future perspective of testing BBB4-G4K as an antibacterial agent. In this regard, according to EUCAST protocols [32], the antibacterial activities must be established by determining the minimum inhibitory concentration (MIC) values after 18–24 h. Consequently, to know the concentration of BBB4 responsible for the MICs that we will observe, it was sufficient to know the amount of BBB4 released at 24 h. The release of BBB4 from BBB4-G4K NPs was biphasic and characterized by an initial fast release in the first few hours, followed by a phase of prolonged sustained release, similar to the release of AMDPC from PEG/PLGA-based micelles [13]. However, while the max AMDPC release was only 77% after 48 h and 60% after 24 h, the BBB4 release after 24 h was practically quantitative (99.5%), thus assuring a higher drug concentration at a lower dose of BBB4-G4K NPs.

Once the CR% curve was obtained, the kinetics and the main mechanisms which govern the release of BBB4 from BBB4-G4K NPs were determined by fitting the CR% curve data with the equations of the zero order model (% cumulative drug release vs. time), first-order model (log% cumulative drug remaining vs. time), Hixson–Crowell model (cube root of % cumulative drug remaining vs. time), Higuchi model (% cumulative drug release vs. square root of time), and Korsmeyer–Peppas model (Ln% cumulative drug release vs. Ln of time) [33,34,35,36]. The highest value of the coefficient of determination (R^2^) of the equations of linear mathematical models was considered as the parameter to determine which model better fits the release data. The R^2^ values were 0.5208 (zero order), 0.8603 (first order), 0.6933 (Korsmeyer–Peppas model), 0.7246 (Hixson–Crowell model), and 0.7232 (Higuchi model), thus establishing the BBB4 release best fitted with the first order kinetic model (Figure 12).

The first-order kinetics model can be expressed by Equation (7):(7)$$\frac{dC}{dt}=-Kc$$
where *K* is the first-order rate constant expressed in units of time ^−1^.

Equation (7) can be in turn expressed according to Equation (8):(8)$$logC_{t}=logC_{0}-\frac{-K_{t}}{2.303}\;$$
where *C_t_* is the concentration of drug release in time *t*, *C*_0_ is the initial concentration of the drug present in the nanocomposite system, *K_t_* is the first-order rate constant, and *t* is the time.

As observable in Figure 12, the first-order kinetic dispersion graph of drug release data was obtained by plotting the log CR% of the drug remaining vs. time that would yield a tendency line with a slope of −*Kt*/2.303 = −0.0556 (*Kt* = 0.128). Since the first-order kinetics model describes the drug release from the system where the release rate of the drug is concentration-dependent, it was established that the BBB4 release from BBB4-G4K is dependent on the residual drug concentration inside the carrier system, and the lower the residual drug concentration, the slower it is.

### 3.10. Dynamic Light Scattering Analysis (DLS)

Table 5 collects the results obtained from DLS analyses on G4K NPs [11] and BBB4-G4K NPs concerning their size (Z-ave, nm), polydispersity index (PDI), and Zeta potential (ζ-*p*).

Figure 13a,b show a representative particle size distribution of the empty dendrimer (G4K) (**a**) [11] and of the BBB4-loaded dendrimer (BBB4-G4K) (**b**), while Figure 13c,d show the ζ-*p* mean distribution of G4K (**c**) [11] and BBB4-G4K (**d**).

For G4K, the mean particle size was 333.4 nm, and the mean PDI was 0.286, whereas for BBB4-G4K, values of 112.1 nm (size) and 0.289 (PDI) were determined. Concerning PDI, the value observed for BBB4-G4K was like that observed for the empty dendrimer G4K, and collectively, the PDI value determined for BBB4-G4K was established for a sufficiently low polydispersity. The particle size of BBB4-loaded NPs was unexpectedly significantly lower than that of the empty dendrimer, contrary to common observations [11,37]. Indeed, the size of the polymer NPs usually increases after the encapsulation of drugs as a function of the amounts of the loaded drug [11,37]. Even if the reported dimensions and DPI of micelles loaded with a pyrazole derivative (AMDPC) recently reported were lower than those obtained by us (68.16 nm and 0.139, respectively) [13], their dimensions were significantly lower than those of the empty PLGA/PEG-PLGA-based polymers reported in the literature, as in our case. In particular, for PLGA/PEG empty polymers, we found in the literature dimensions of 250–360 nm [37] and of 125.8 nm [38]. For the good functioning of a nano delivery system, in addition to morphology, the size of NPs plays a pivotal role and influences the formulation distribution, cytotoxicity, targeting ability, and drug release profile [30]. Biomedical applications require sizes lower than 200 nm, with an optimal size of 100–200 nm. Particles larger than 200 nm tend to activate the lymphatic system and are removed from circulation quicker [30], while NPs with a size of about 100 nm, as that determined for BBB4-G4K NPs, have a larger surface area to volume ratio and are more effective and faster. Concerning ζ-*p*, while the ζ-*p* value reported for AMDPC-loaded micelles, which were made of neutral PEG-PLGA polymers, and the pyrazole derivative AMDPC was negative (−16.87 mV), BBB4-G4K NPs maintained a positive ζ-*p* value (although halved in absolute value) of +28.9, due to the cationic structure of the reservoir dendrimer G4K having a ζ-*p* of +66.1 mV. Importantly, note that a high ζ-*p* value around ±30 mV, as that observed for BBB4-G4K NPs, typically assures good physical stability of the formulation in water solution with no tendency to form aggregates [39].

### 3.11. Potentiometric Titrations

The prepared BBB4-G4K NPs, like the reservoir dendrimer used to prepare them, did not have quaternary and permanently protonated ammonium groups but were both characterized by having primary amine groups whose protonation is reversible, depending on the pH value of the environment. The protonation of amine groups of NPs intended for developing new antibacterial agents, as in our future perspectives for BBB4-G4K NPs, is essential for having a positively charged surface, which is crucial to interacting with the surface of bacteria and providing significant antibacterial activity. Therefore, we considered it important to know the pH values at which BBB4-G4K can be protonated and mainly if it will be protonated in the physiological pH range of 4.5–7.5. To obtain this information, the potentiometric titration of BBB4-G4K and, for comparative purposes, that of G4K has been carried out according to Benns et al. [21].

The measured pH values were reported in the graph vs. the aliquots of HCl 0.1 N added, obtaining the titration curves of G4K and of BBB4-G4K (Figure 14, yellow and red lines, respectively). Subsequently, from the titration data, the dpH/dV values were computed and reported in the graph vs. those of the corresponding volumes of HCl 0.1N, obtaining the first derivative lines of the titration curves (Figure 14, light blue and dark blue lines, respectively). The maxima of the first derivative curves corresponded to the volumes of HCl necessary to have G4K and BBB4-G4K in the protonated forms. Interestingly, for both samples, two maxima were observed due to the presence of two different types of primary amine groups of lysine residues (^α^NH_2_ and ^ε^NH_2_), thus establishing the existence of a two-step protonation process. According to the obtained results, the two protonation steps of BBB4-G4K occurring at pH = 6.85 and 4.80 in the physiological pH range (4.8–8, depending on whether it is measured), it should be completely protonated, also assuring in vivo applications and a cationic chemical structure capable of strong electrostatic interactions with a negatively charged bacterial surface.

In contrast, G4K was firstly protonated at pH = 4.85 and secondly at a very low pH value of 2.00. This finding also explains why G4K, when tested for its antibacterial properties in Mueller Hinton Broth (Dehydrated) (pH = 7.4), was completely inactive against all bacteria species assayed (not reported results).

## 4. Conclusions and Future Perspectives

In this study, with the future perspective of developing a new water-soluble pyrazole-based in vivo administrable antibacterial agent, we physically entrapped the pyrazole derivative BBB4 in a biodegradable and non-cytotoxic cationic dendrimer that we synthetized, thus obtaining water-soluble dendrimer NPs loaded with BBB4 (namely BBB4-G4K NPs). A scrupulous characterization to determine the chemical composition of BBB4-G4K NPs was performed by acquiring FTIR spectra, also processed by PCA, and NMR analysis, which confirmed the success of the encapsulation. In particular, the ^1^H NMR spectrum of BBB4-G4K was useful for determining the moles of the BBB4 loaded per mole of G4K and for computing the MW of the complex, which was in accordance with that determined by considering the drug loading (DL%) value obtained by HPLC analyses. The DL% was 22.5-fold higher than that of NPs obtained by encapsulating a pyrazole derivative in commercial copolymers (PEG-PLGA), which was recently reported. The release experiments showed a biphasic release profile governed by first-order kinetics and a quantitative release after 24 h which, in future microbiological experiments to determine MICs, will assure a high concentration of BBB4 at a low dosage of BBB4-G4K NPs. From DLS experiments, it was established that the BBB4-G4K particles were nanosized, with dimensions assuring low systemic toxicity, while micrographs obtained by SEM showed a spherical morphology which includes a high surface area, which typically translates to a high systemic residence time and bio-efficiency. PDI was low, and the ζ-*p* value was positive and sufficiently high to ensure a high stability in an aqueous solution and no tendency to form aggregates. Titration experiments demonstrated the presence of nitrogen atoms which proved to remain protonated in the whole physiological pH range.

Mostly, the overall merit of the present work consists of having prepared a BBB4 formulation with a water-solubility 105-fold higher than that of untreated BBB4, without resorting to the use of co-solvents (PEG), surfactants, stabilizers, or emulsifiers, which can be dangerous for humans. In particular, by nanotechnological manipulation using dendrimers, the water insoluble BBB4, devoid of scientific relevance justifying further research, as it is not clinically applicable, has been converted to a form worthy of further biological investigations, such as the evaluation of its antibacterial properties and of its cytotoxicity, which will be the subject of our next work.

## Data Availability

All data concerning this study are contained in the present manuscript or in previous articles whose references have been provided.

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
