# Peer review of "Preparation and Physicochemical Characterization of Water-Soluble Pyrazole-Based Nanoparticles by Dendrimer Encapsulation of an Insoluble Bioactive Pyrazole Derivative"

_nanomaterials, 2021, doi:10.3390/nano11102662_

Round 1

Reviewer 1 Report

The proposed manuscript is interesting and I suggest its publication after some corrections.

  1. In the Results and Discussion sections, there are some information that according to my judgement would fit better in the Experimental section; i.e. 3.4. Preparation of BBB4-Loaded Dendrimer NPs (BBB4-G4K). Besides, authors should consider if there are some information in the sections 3.6.1. BBB4 calibration curve and 3.6.2. Determination of BBB4 concentration in the prepared BBB4-G4K NPs, DL% and EE% that would be more suitable in the Experimental section as well.
  2. Check Figure 7. There are some lines in the body of the figure that meaning is unclear.

Author Response

The proposed manuscript is interesting, and I suggest its publication after some corrections.

We thank the Reviewer for his positive comment and for having appreciated our study.

In the Results and Discussion sections, there are some information that according to my judgement would fit better in the Experimental section; i.e. 3.4. Preparation of BBB4-Loaded Dendrimer NPs (BBB4-G4K). Besides, authors should consider if there are some information in the sections 3.6.1. BBB4 calibration curve and 3.6.2. Determination of BBB4 concentration in the prepared BBB4-G4K NPs, DL% and EE% that would be more suitable in the Experimental section as well.

We thank the Reviewer for his suggestions, and, even if we have not changed the location in the manuscript of the paragraphs indicated by the Reviewer, we have changed their titles, for more clarity and to address his concerns. Particularly, concerning paragraph 3.4. Preparation of BBB4-Loaded Dendrimer NPs (BBB4-G4K), we make kindly note to the Reviewer, that in Experimental Section, there is already a paragraph reporting the experimental procedure performed to prepare BBB4-G4K NPs, containing all experimental details useful to reproduce BBB4-G4K preparation [namely 2.4. Preparation of BBB4-Loaded Dendrimer NPs (BBB4-G4K)]. On the contrary, the paragraph indicated by the Reviewer (3.4) does not contain the experimental procedure, but a discussion concerning the experimental procedure adopted by us to prepare the BBB4-loaded NPs, the motivations for our choice (completed with the relative references), and the results achieved by applying such method. So, we think that the original location of the paragraph 3.4. in Results and Discussion Section could be correct. However, to avoid confusion and address the Reviewer concern, in the revised version of our manuscript, we have differentiated the titles of the paragraphs involved in this question. Please, see lines 218, and 420-421.  Similarly, it can be observed for paragraphs 3.6.1 and 3.6.2. Indeed, in Experimental Section, two paragraphs (2.7.1. and 2.7.2.), already exist, which contain the experimental procedure, the experimental details, and details about the instruments and experimental conditions, concerning the construction of the BBB4 calibration model (2.7.1.) and the determination of BBB4 content in BBB4-G4K NPs, of DL% and of EE% (with related equations) (2.7.2.). On the contrary, paragraph 3.6.1. and 3.6.2. contain the relative results and their discussion, so, we think that their inclusion in the Results and Discussion Section is appropriate. Anyway, for more clarity and satisfy the Reviewer, the titles of paragraphs 3.6.1. and 3.6.2. have been slightly changed. Please, see line 554 and lines 580-581.

Check Figure 7. There are some lines in the body of the figure that meaning is unclear.

We thank the Reviewer for his comment which denote its accuracy in reading our manuscript. Thus, since all signals in the spectrum (Figure 7), were attributed already in the original version of the manuscript, we think that the Reviewer with “some lines” referred to the dark red curves present in the spectrum. Well, we explain that such lines are the integrals belonging to the signal/signals which it covers and indicate the number of proton atoms which contribute to give that signal/signals. For more clarity, an explanation has been included in the Figure 7 caption, which has been slightly modified. Please see lines 536-539.

Reviewer 2 Report

The article presents the synthesis and characterization of water-soluble pyrazole-based nanoparticles created from encapsulation of an insoluble pyrazole derivative into dendrimers. Such topic is new and promising because in the latest publications the attempts are made to encapsulate bioactive insoluble drugs into dendrimers and other nanoparticles to improve their biocompatibility. The article is well written and easy to read and understand. The huge number of experiments was made to characterize the nanoparticles. The results are correctly done and well described. The conclusions are supported by the data. As for English, for me (non-native) it is O.K.  I did not find any serious remarks to mention. In general, I think the article can be published after small corrections.

I have only minor remarks:

1) In fig. 12 is clearly seen that the kinetics can be approximated by second order kinetic model.

2) It would be perfect to add results in present article on cytotoxicity of BBB4 on normal cells (hu-394 man keratinocytes HaCaT) (as authors mentioned).

Author Response

The article presents the synthesis and characterization of water-soluble pyrazole-based nanoparticles created from encapsulation of an insoluble pyrazole derivative into dendrimers. Such topic is new and promising because in the latest publications the attempts are made to encapsulate bioactive insoluble drugs into dendrimers and other nanoparticles to improve their biocompatibility. The article is well written and easy to read and understand. The huge number of experiments was made to characterize the nanoparticles. The results are correctly done and well described. The conclusions are supported by the data. As for English, for me (non-native) it is O.K.  I did not find any serious remarks to mention. In general, I think the article can be published after small corrections.

We are very grateful to the Reviewer for his positive comments and for having appreciate our study.

I have only minor remarks:

  • In fig. 12 is clearly seen that the kinetics can be approximated by second order kinetic model.

We apologize in advance with the Reviewer, but his comment is not clear. Why does he assert that from Figure 12, the kinetic model that fit the cumulative release profile of BBB4 (reported in Figure 11) is the “second order kinetic model”? As we reported in the main text, we fitted the most common mathematical models used to explain the mechanisms which govern the release profiles of drugs from nanoparticles [zero order model (% cumulative drug release vs time), first-order model (log % cumulative drug remaining vs time), Hixson-Crowell model (cube root of % cumulative drug remaining vs time), Higuchi model (% cumulative drug release vs square root of time), and Korsmeyer-Peppas model (Ln % cumulative drug release vs Ln of time) [Ref. 33-36], lines 678-681)]. Accordingly, Figure 12 represents the dispersion graph obtained fitting the first kinetic model to the data of release profile and the relative linear regression whose R2 was the highest among those obtained by fitting other kinetic models (please, see lines 681-686), thus establishing that the first order kinetic model is the mathematical model that better explains the BBB4 release profile. Consequently, the assertion of the Reviewer in understandable. In addition, as reported in a remarkable review on the drugs release profiles and the possible mathematical kinetic models (Review Article: M. PADMAA PAARAKH, PREETHY ANI JOSE, CM SETTY, G.V. PETER CHRISTOPER. RELEASE KINETICS – CONCEPTS AND APPLICATIONS. International Journal of Pharmacy Research & Technology | Volume - 8 -2018), many are the appliable possible kinetic models, but the second order kinetic model is never mentioned.  Moreover, as reported in other many articles, the second order kinetic model usually fits the adsorptive removal of substances from loaded materials by opportune adsorbents as carbonaceous materials or bentonites. Please, consider:

Asmita Shrestha1, Bhoj Raj Poudel2, Manoj Silwal1, Megh Raj Pokhrel. ADSORPTIVE REMOVAL OF PHOSPHATE ONTO IRON LOADED LITCHI CHINENSIS SEED WASTE. Journal of Institute of Science and Technology 2018, 23: 81-87. DOI: https://doi.org/10.3126/jist.v23i1.22200.

Luo, B., Huang, G., Yao, Y. et al. Comprehensive evaluation of adsorption performances of carbonaceous materials for sulfonamide antibiotics removal. Environ Sci Pollut Res 28, 2400–2414 (2021). https://doi.org/10.1007/s11356-020-10612-7.

Bourliva, A., Michailidis, K., Sikalidis, C., Filippidis, A., & Betsiou, M. (2013). Lead removal from aqueous solutions by natural Greek bentonites. Clay Minerals, 48(05), 771–787. doi:10.1180/claymin.2013.048.5.09

2) It would be perfect to add results in present article on cytotoxicity of BBB4 on normal cells (hu-394 man keratinocytes HaCaT) (as authors mentioned).

We appreciated that the Reviewer think that by the addition of results concerning cytotoxicity of BBB4, our work could be “perfect”, but as reported in many points of the present manuscript, the biological evaluations of both BBB4 and BBB4-G4NPs (including cytotoxicity data of BBB4 on human normal cells) were not in the scope of this study. Anyway, we have reported data concerning the cytotoxicity of the empty dendrimer adopted to entrap and solubilize BBB4, because we retained not rational starting from a delivery system which could be cytotoxic. The scope of the present manuscript was synthetizing and characterizing water-soluble BB4-loaded nanoparticles, by encapsulating BBB4 in a not cytotoxic carrier (data reported) for make it potentially in vivo administrable. We considered premature and meaningless assessing eventual bioactivities of a molecules which in its free form is insoluble and not suitable for biomedical application. Moreover, we think that there would be no sufficient space in a study already quite complex like this, for additional biological tests, that in our opinion require a separate and dedicated study, to make a good work.